# Attention amplifies neural representations of changes in sensory input at the expense of perceptual accuracy

Vahid Mehrpour [1,2,3✉], Julio C. Martinez-Trujillo [4] & Stefan Treue [1,2,3,5✉]

Attention enhances the neural representations of behaviorally relevant stimuli, typically by a push–pull increase of the neuronal response gain to attended vs. unattended stimuli. This selectively improves perception and consequently behavioral performance. However, to enhance the detectability of stimulus changes, attention might also distort neural representations, compromising accurate stimulus representation. We test this hypothesis by recording neural responses in the visual cortex of rhesus monkeys during a motion direction change detection task. We find that attention indeed amplifies the neural representation of direction changes, beyond a similar effect of adaptation. We further show that humans overestimate such direction changes, providing a perceptual correlate of our neurophysiological observations. Our results demonstrate that attention distorts the neural representations of abrupt sensory changes and consequently perceptual accuracy. This likely represents an evolutionary adaptive mechanism that allows sensory systems to flexibly forgo accurate representation of stimulus features to improve the encoding of stimulus change.

[1] Cognitive Neuroscience Laboratory, German Primate Center - Leibniz Institute for Primate Research, Göttingen 37077, Germany. [2] Bernstein Center for Computational Neuroscience, Göttingen, Germany. [3] Leibniz ScienceCampus Primate Cognition, Göttingen, Germany. [4] Department of Physiology and Pharmacology, University of Western Ontario, London, ON, Canada N6A 5B7. [5] Faculty for Biology and Psychology, University of Göttingen, Göttingen, Germany. [6] Present address: Department of Psychology, University of Pennsylvania, Philadelphia, PA 19104, USA. ✉email: v.mehrpour@gmail.com; treue@gwdg.de

The torrent of information entering high-performance sensory systems, such as the visual system of primates, mostly represents steady-state aspects of the environment, i.e., spatially or temporally unchanging input. Any deviation in the input usually reflects highly informative and relevant events. The neural representation and perception of such change events provide an excellent setting to evaluate the powerful role of attention in shaping perception. Attention has been shown to enhance the neural representation of behaviorally relevant over irrelevant sensory inputs[1–3]. It does so in a carefully orchestrated up- and down-regulation of response gain across populations of neurons encoding attended and unattended stimuli[4–8]. Attention has also been shown to improve the perception of attended stimuli and consequently behavioral performance—likely by reducing noise, strengthening signals, shortening response latency, optimizing inter-areal communication, and keeping track of relevant stimuli across eye movements[9–20]. In most contexts, the enhanced neural representation of attended stimuli goes hand-in-hand with improved perception and behavioral benefit—e.g., in discriminating a feature of a stationary stimulus[21] or in a visual search task[22,23]. By contrast, for the detection of abrupt changes in the environment, an enhanced representation of a change may benefit behavior but would cause a misperception of its magnitude. Here we disassociate these two potentially conflicting aims of attentional systems: the improvement of perception by an enhanced accuracy of the attended stimulus representation vs. the improvement of behavioral performance by a magnification of task-relevant sensory aspects, even if that entails creating a "less veridical" representation of the stimulus.

In order to determine whether attention acts to forgo accuracy in the representation of abrupt changes in the environment, we recorded single-cell activity from 52 neurons in the middle temporal area (MT) of two rhesus monkeys, while the animals performed a motion direction change detection task. We isolated the impact of attention on the neural representation of such abrupt direction changes by comparing representations of attended vs unattended changes. Our results show that attention distorts the representations of motion direction changes. While the distorted representation results in neural misrepresentations and misperceptions of the change events, it increases their perceptual detectability and, consequently, improves the behavioral performance in the change detection task.

## Results

**Attention effect on the neural representation of changes.** To evaluate the effects of voluntary attention on the accuracy of the neural representation of the magnitude of abrupt changes we used a well-established spatial attention paradigm[6,7,24,25]. Two rhesus monkeys were trained to perform a direction change detection task. On each trial, two random dot patterns (RDPs) moved in 1 of 12 evenly spaced directions. After a random interval (300–4000 ms) a clockwise direction change of 25° (see Supplementary Note 1 for the sign convention used in this study) occurred in one of the RDPs, lasting 200 ms before the motion returned to its initial direction. In randomly interleaved trials, the monkeys were cued to covertly attend to either the RDP inside the receptive field (RF) or to the other RDP outside the RF. The monkeys were rewarded for detecting a direction change of the cued stimulus within a reaction time window from 250 to 700 ms after the change onset (Fig. 1a; see Methods for details). Given that the stimulus change is brief and the following reaction time window is also short, the task is challenging, with the animals' performance well above chance, but far from saturated (mean ± SEM = 87% ± 1% for monkey M; 87% ± 2% for monkey F). This paradigm allowed us to compare the responses of each

neuron (and its encoding accuracy) to an attended vs an unattended direction change.

To visualize the results, we consider our data set as representative for the population of MT neurons responding to a given motion stimulus and its change. Given this assumption we determined the temporal evolution of the activity profile of the MT population throughout the trials (Fig. 1b–e; see Methods for details). Figure 1b–c depicts the MT population response to an attended (Fig. 1b) and an unattended (Fig. 1c) stimulus. As expected, both profiles show a shift in the activity peak following the direction changes of the stimulus in the RF. In similar agreement with the known attentional gain modulation of visual responses in extrastriate cortex, the MT responses are enhanced for the attended vs. the unattended stimulus inside the RF. However, surprisingly, even just a visual inspection of the profiles (see zoomed-in panels Fig. 1d–e; black line represents the encoded stimulus direction; white line represents the physical stimulus) reveals an overrepresentation of the stimulus direction after the initial stimulus change, which is more pronounced when the stimulus is attended.

To quantify the effect of attention on the accuracy of the encoded direction change, we averaged the pre- and post-change population responses from −700 to 0 ms (pre-change) and from 100 to 200 ms (post-change) (Fig. 2a). Pre- and post-change responses to the attended stimulus show a multiplicative attentional enhancement of about 13% compared to the responses to the unattended stimulus (Supplementary Note 2). Most importantly, and in line with the visual inspection of the time courses in Fig. 1d–e, the physical direction change of 25° is represented in the post-change population response as a 39° change, an overestimation of 14°. Evaluation of the encoding accuracy of an unattended stimulus shows a similar but markedly smaller misrepresentation (Fig. 2a), most likely reflecting the effects of sensory adaptation[26]. The comparison of the attended and unattended representation shows that attention enhances, rather than reduces, a perceptual inaccuracy present in the neural representation of the magnitude of unattended stimulus changes.

As a further quantification of our observations we separately determined pre- and post-change direction tuning curves for each cell by fitting its responses (averaged across the same time windows used above) with von Mises functions (see Methods for details; see Supplementary Note 3 for direction tuning curves of an example MT cell as well as fit quality and direction selectivity distributions for the cells). The distributions of the individual cell's preferred direction shift in both attentional conditions confirm our previous analysis (Fig. 2b). The median shift for an attended stimulus is −14° ($p = 0.0005$, two-sided Wilcoxon's signed-rank test for distribution with zero median), whereas it is −6° ($p = 0.0000005$, two-sided Wilcoxon's signed-rank test for distribution with zero median) for an unattended stimulus (Fig. 2b). The difference between tuning shifts in attended and unattended conditions is highly significant ($p = 0.001$, paired two-sided Wilcoxon's signed-rank test). Our results remained consistent when parsed by the data collected from each animal individually (Supplementary Note 4).

In a series of analyses, we confirmed that our results are independent of the standard deviation of the Gaussian kernel used to calculate the spike density function (SDF), the tuning characteristics of neurons, the position of the pre- and post-change spike count windows, and the symmetry of the function used to fit the tuning data (Supplementary Note 5).

We also employed a simple linking model to translate the shift of the direction tuning of MT cells induced by the attended and unattended direction changes into an estimate of the perceived abrupt direction change. According to a labeled-line model, each possible direction is encoded by one neuron (or neuronal

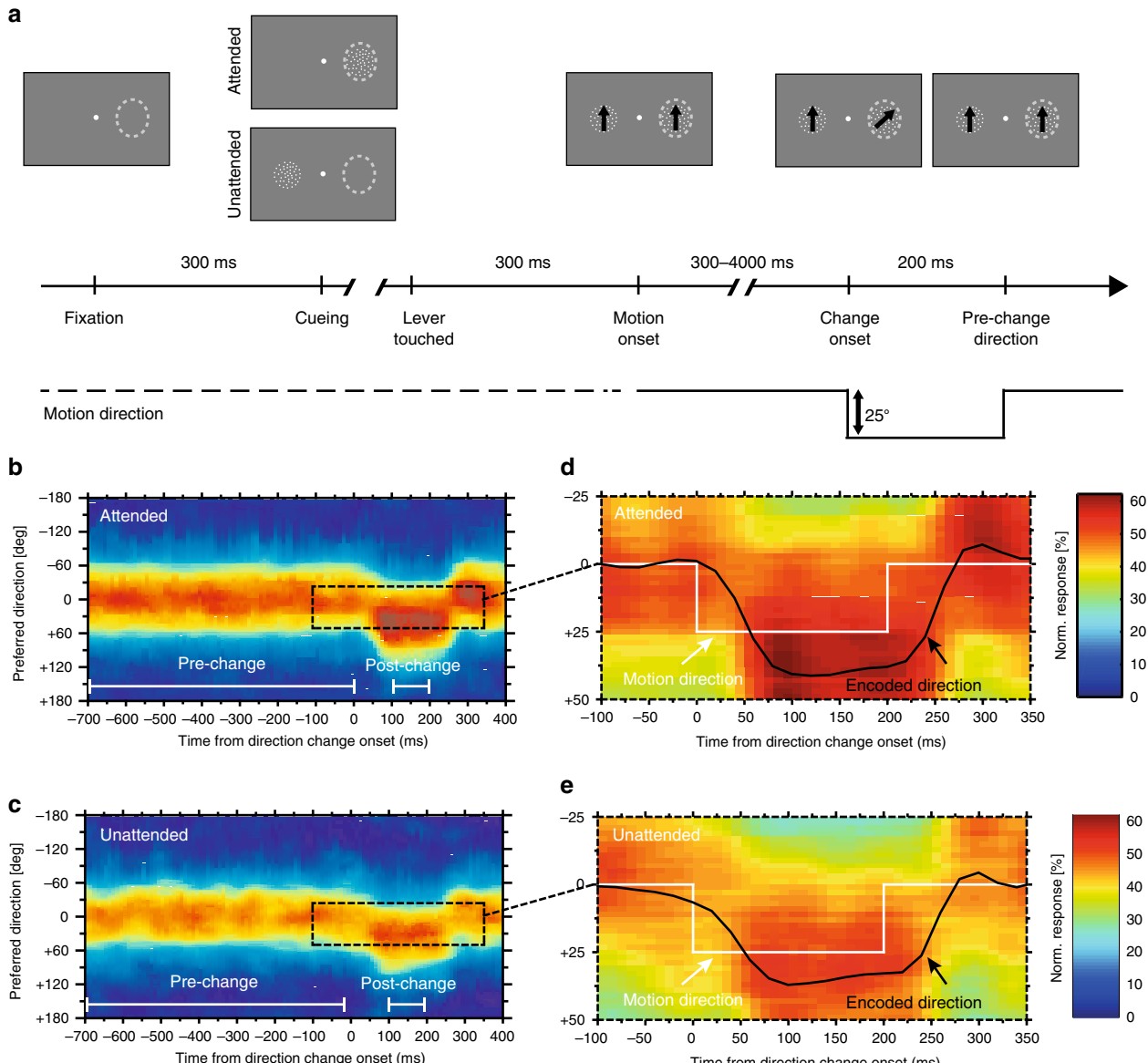

**Fig. 1 Neural representation of attended and unattended direction changes in MT. a** Motion direction change detection task. A trial began with the appearance of a central fixation point (indicated by a white dot). After the monkey foveated the fixation point, a static RDP was shown either inside the neuron's receptive field (RF, dashed ellipses) or in the hemifield opposite to the RF, cueing the animal to the location of the upcoming target. The trial was continued if the monkey touched a lever. 300 ms after the lever was touched, another RDP was displayed at the other location and both RDPs began to move in 1 of 12 evenly spaced directions within their stationary apertures. Clockwise direction changes of +25° could occur in either the target or distractor in a time interval from 300 to 4000 ms after the motion onset. The stimulus motion was returned to its pre-change direction after 200 ms. To receive a fluid reward the animal was required to detect the change in the target by releasing the lever and to ignore a similar change in the distractor. Each trial belonged to one of two spatial attention conditions: target inside the RF (attended) or distractor inside the RF (unattended). **b–e** Visualization of MT population response profiles in the attended (**b**, **d**) and unattended (**c**, **e**) conditions. The x-axis plots time relative to the direction change onset, the y-axis indicates the preferred direction of a given subpopulation of neurons, and the magnitude of responses is color-coded according to the color scale shown on the right. Areas enclosed by the dashed rectangles in the left graphs are magnified in the right graphs. The white line in the right graphs indicates the time course of the direction of the stimulus, the black curve plots the encoded direction determined by fitting the population response with a von Mises function every 20 ms (see Methods for details). Source data are provided as a Source Data file.

subpopulation). Each neuron's "label" is its pre-change preferred direction and the population response profile is the distribution of responses across all neurons to a given stimulus (Supplementary Note 6). Based on a winner-take-all readout approach, the perceived (readout) direction is the preferred direction (label) of the neuron with the largest evoked response to the stimulus, i.e., the location of the peak of the population response profile[26,27]. This simple model predicts that direction tuning shifts induced by the attended and unattended abrupt direction changes result in

the overestimation of the perceived direction change with the magnitudes we saw in the context of Fig. 2a (Supplementary Note 6).

**Perceptual representation of attended change events.** Given the central and causal role of area MT for visual motion perception[28,29], we wondered whether the perceptual representation of the abrupt direction change also follows its neural representation. To test this, we examined the perception of

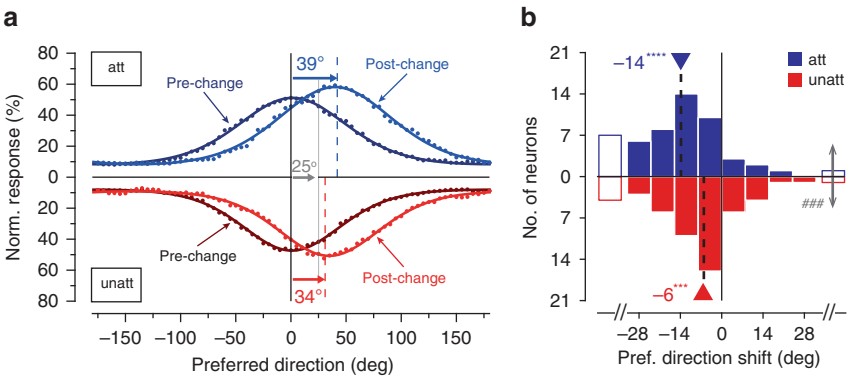

**Fig. 2 Quantification of attention effect on MT neural responses. a** Population response curves: population responses (Fig. 1b–c) averaged across a pre-change time interval from −700 ms to 0 ms (black points) and a post-change interval from +100 to +200 ms (colored points) in the attended (top panel) and unattended (bottom panel) condition. In each attention condition we fitted these time-averaged, pre- and post-change population responses with von Mises functions (solid curves). The dashed lines in each attentional condition indicate the location of the population's peak response following the direction change of 25°. Error in the peak location obtained from the bootstrap procedure was 0.1° for pre-change (attended), 0.6° for post-change (attended), 0.2° for pre-change (unattended), and 0.4° for post-change (unattended) (see Methods for details). **b** Distribution of direction tuning curve shifts caused by attended (blue) and unattended (red) direction changes of +25° for a population of MT units (n = 52 units). The median values and the corresponding p values are labeled (***p = 0.0005; ****p = 0.0000004 two-sided Wilcoxon's signed-rank test for distribution with a median of 0; ###p = 0.001 paired two-sided Wilcoxon's signed-rank test). Unfilled bars represent units for which the magnitude of tuning shift was greater than 40°. The sign convention is that positive and negative values denote clockwise and counterclockwise displacements, respectively (Supplementary Note 1). Source data are provided as a Source Data file.

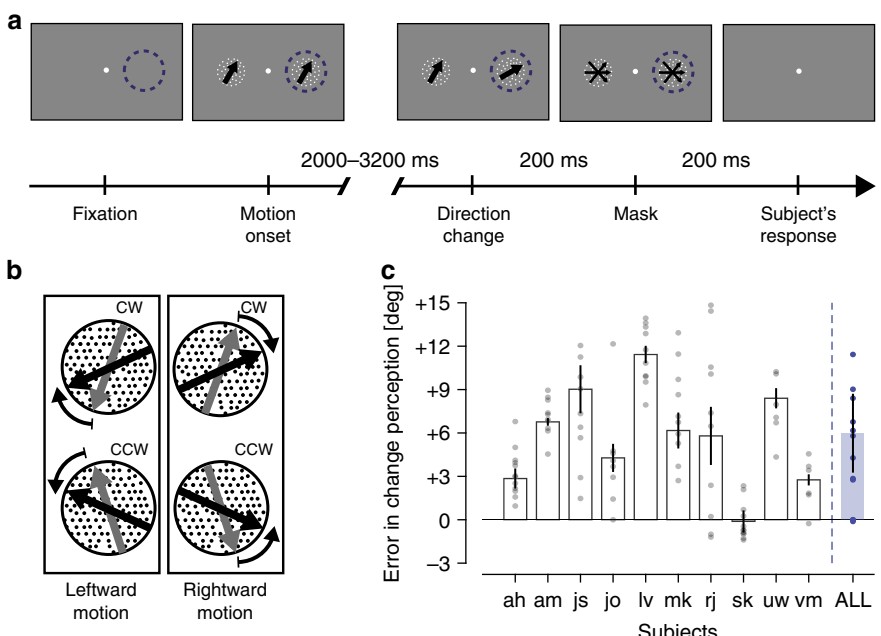

**Fig. 3 Quantifying the perceived magnitude of attended direction changes in humans. a** Subjects had to foveate a central fixation point (white dot), covertly attend to the right side of the screen (dashed circle) and press a gamepad button to start a trial. Then, two coherent RDPs moving in the same direction were displayed. At a random time between 2000 and 3200 ms after motion onset, a direction change of 22°, 25°, or 27° occurred in the right stimulus. 200 ms later, the RDPs were replaced by two mask RDPs displayed for 200 ms. The subjects had to report whether the post-change motion direction was upward or downward from horizontal (see Methods). **b** Stimulus configuration in the psychophysical task. In a given trial, RDPs could move either leftward or rightward. For each motion direction, one of two clockwise (CW) and counterclockwise (CCW) direction changes could occur. Gray and black arrows indicate pre- and post-change motion directions, respectively. **c** Error in direction change perception: subject codes are shown along the abscissa (last bar: median effect across all n = 10 subjects) and the ordinate represents the median error in the perceived direction change. According to our sign convention, positive and negative errors along the y-axis correspond to an overestimation and underestimation of a direction change, respectively. Filled circles show the data points underlying each bar in the graph and error bars indicate the median absolute deviations. Source data are provided as a Source Data file.

attended direction changes in human subjects using a design similar to the one of our monkey electrophysiology study (Fig. 3a). Human subjects were instructed to maintain their gaze on a central fixation spot on a computer display. On a given trial, two RDPs moving in the same direction (rightward or leftward, with some deviation off the horizontal) were presented to the left and right of the fixation spot and subjects were instructed to attend to the right RDP. At a random time point the direction of

the attended RDP changed by 22°, 25°, or 27° towards the horizontal (Fig. 3b). Using a staircase approach, we adjusted the pre-change direction such that the perceived post-change direction was aligned with the horizontal direction (see Methods for details). This counterbalanced design allowed us to estimate the subject's perception of the magnitude of the attended direction change, unbiased by known issues, such as direction repulsion[30,31]. It is notewothy that it was not our intention to isolate the effect of attention on the perception of behaviorally relevant direction changes. Rather, we aimed to demonstrate that (assuming a similar neural misrepresentation in humans and our monkeys) humans experience a perceptual overestimation of the magnitude of attended direction changes.

We found that all but one subject overestimated the post-change direction (Fig. 3c), with an average overestimation of 6°. Although the magnitude of the effect is smaller than the overrepresentation in MT, it is highly significant ($p = 0.004$, two-sided Wilcoxon's signed-rank test for distribution with zero median), supporting the hypothesis that the misrepresentation we observed in MT is the neural basis for the perceptual overestimation in our human subjects.

## Discussion

The visual system of primates operates in a predominantly stable, steady-state sensory environment. Correspondingly, changes in the sensory input are highly informative and the visual system has evolved to be particularly sensitive to these signals[32,33]. Here we investigated how attention impacts the neural representation of such change events. This offers an opportunity to dissociate two potentially conflicting core conceptual aims of attentional systems: the improvement of perception by an enhanced dominance and accuracy of attended stimulus representations vs. the improvement of behavioral performance by a magnification of task-relevant sensory aspects, even if that entails creating a "less veridical" stimulus representation.

The model system for our study was the neural basis of the effects of attention on the perception of abrupt changes in the direction of a moving stimulus. We determined putative population responses to such changes from single-cell recordings in extrastriate area MT of two rhesus monkeys performing a motion direction change detection task. Area MT is known to play a critical role in the perception of visual motion[29,34–36] and it contains a large proportion of direction-selective cells whose responses are modulated by attention[1,2,5,6].

Our results show that a behaviorally relevant change in the direction of motion induces a shift in the center of MT population response that is more than 50% larger than predicted by the physical change in the stimulus. The psychophysical results of our direction discrimination experiment in human complement the physiological effects by demonstrating a similar perceptual overestimation of direction changes. These results strongly argue for a role of attention in overrepresenting sensory changes in our environment, particularly if the change per se is behaviorally relevant. It is noteworthy that our human psychophysical experiment, while invoking attention, does not distinguish between the effects of attention and adaptation.

One contributor to the overestimated neural representation of the direction change might be changes in the tuning of MT neurons after motion adaptation[26]. This could explain the shifts in the population response to an irrelevant stimulus, found in our unattended condition (sensory shifts). However, in our attended condition the overrepresentation was much larger, despite identical sensory stimulation. Could an increased adaptation, caused by the attentional increase of responses cause this effect? Our data argue against this. If the increase in the tuning shifts associated

with attention (attentional shifts, attended shift–unattended shift) is caused by the attentional modulation of pre-change responses, we would expect that neurons with larger attentional modulation of their responses exhibit larger attentional shifts. Instead, we found that there is no correlation between the attentional modulation of pre-change tuning parameters and the attentional shifts (Pearson's correlation: $r < 0.2$, $p > 0.05$). To further investigate the interaction between attention and pre-change visual motion adaptation, we examined the relationship between the decay of responses over the course of visual motion exposure prior to the direction change as a measure of adaptation and the attentional shift and did not find any correlation between them (Pearson's correlation: $r = 0.04$, $p = 0.8$; Supplementary Fig. 7a). Moreover, if attention increases the adaptation-induced tuning shifts simply by modulating the pre-change responses, we would expect that the cells with larger negative sensory shifts in unattended condition (which presumably underwent stronger adaptation over the course of pre-change visual motion exposure) exhibit larger negative attentional shifts. In other words, there should be a positive correlation between a shift in the sensory (unattended) condition and the attentional shift. Our analysis (Supplementary Fig. 7b) shows an opposite relationship (negative correlation, Pearson's $r = -0.81$, $p = 2 \times 10^{-13}$): cells with large negative tuning shifts in unattended condition were those either not affected by attention, or even showed positive attentional shifts (opposite to the sign of main effect) and vice versa (i.e., the cells with positive sensory shifts had the largest negative attentional shifts). This suggests that attention does not increase the sensory shifts simply by modulating the pre-change responses, but rather combines with the effect of adaptation to improve the neural representation of the change event. All together, these results demonstrate that while adaptation might cause the misrepresentation in the unattended condition, the larger effects in the attended condition are not the direct results of an attentionally enhanced adaptation, but rather reflect attention, like adaptation, causes an overrepresentation of motion change. Similarly, psychophysical studies in human subjects demonstrate that adaptation does not modulate the magnitude of attentional effects on performance[37].

Normalization mechanisms have been used to explain the effects of adaptation[38] and attention[39,40], and they offer a plausible explanation for our electrophysiological findings. These mechanisms assume that visual motion exposure not only suppresses the feedforward drive of neurons but also weakens the activity of the neurons that make up the normalization pools, resulting in a decrease in the suppressive drive and therefore response facilitation[38]. A normalization model that incorporates both adaptation and attention might be able to account for the effects induced by unattended and attended direction changes. Based on this model, direction change overestimation in MT results from the suppression of the feedforward drive as well as a weakened normalization following the visual motion exposure. Attention might increase the overestimation by modulating both components.

A human psychophysics study[19] has shown that an attentional enhancement of spatial resolution leads to an enhanced perpetual performance for peripheral vision, but a reduced performance at central retinal locations. Our study complements this rare example of an attentional modulation that reduced accurate representation of stimulus. We showed a reduced neural and perceptual accuracy in motion directions after abrupt direction changes.

Interestingly, recent studies have used a signal detection approach to show that attention increases the sensitivity of neurons in primate prefrontal cortex to discriminate between the neural representations of two stimuli[41–43]. Our results extend this

finding: we observe an enhanced separation between the responses of a whole population of MT neurons to a stimulus before and after a given direction change, increasing this signal-to-noise relationship, supporting an enhanced sensitivity for the change.

To summarize, we show that, in a motion direction change detection task, attention distorts the neural representation of the change, increasing its perceptual detectability, but causing a misperception of the direction change magnitude. In other words, our study dissociates the widely documented attentional improvements of perception from attentional improvements of behavioral performance by showing that under appropriate circumstances visual attention mechanisms enhance the perceptual saliency of abrupt changes in sensory signals at the expense of an accurate representation of the physical change. This appears to be a highly adaptive and dynamic feature of an attentional system to produce neural representations of sensory events that optimize behavioral outcomes.

## Methods

**Monkey electrophysiological study**. Research with non-human primates represents a small but indispensable component of neuroscience research. The scientists in this study are aware and are committed to the responsibility they have in ensuring the best possible science with the least possible harm to the animals[44].

**Experimental procedures**. Data were collected from two male rhesus macaques. After initial training of monkeys, they underwent surgery under anesthesia under appropriate anesthesia and with appropriate peri-surgical analgesia to implant a headpost, a scleral eye coil, and a recording chamber. Animal procedures complied with all applicable German and European regulations, as well as the NIH Guide for Care and Use of Laboratory Animals and were approved by the responsible regional government office.

A custom-written computer program running on an Apple Macintosh PowerPC controlled the stimulus presentations and recorded the behavior of the animal and monitored eye position.

**Visual stimuli**. RDPs of bright dots of the luminance of 55 cd m$^{-2}$ were displayed on a dark monitor screen of 0.1 cd m$^{-2}$, placed 57 cm away from the animal. Dots were randomly plotted with a density of 5 dots per degree square within a virtual circle matching the size of the classical RF of the neuron under study. The speed of all the dots was uniform over the entire RDP and was set to the preferred speed of the neuron. In each trial, dots coherently moved in 1 of 12 equally spaced directions. Clockwise direction changes of 25° occurred at random time points between 300 and 4000 ms after the onset of visual motion. Two hundred millisecond after each clockwise direction change, a second counterclockwise change returned the motion direction to its initial value prior to the change.

**Electrophysiological recording**. Extracellular single-unit recordings were made from two hemispheres of two awake, behaving rhesus monkeys seated in a primate chair with the head restrained. The neural responses were recorded from 52 isolated MT cells (40 neurons from monkey M and 12 from monkey F) with tungsten microelectrodes (impedance 0.5–2 Ω, Microprobe and FHC). MT was identified by its anatomical location, a high percentage of direction-selective cells, and the eccentricity and size of the RFs.

**Behavioral task**. Monkeys were trained to perform a motion direction change detection task. Each trial began with a fixation point displayed in the center of a blank computer screen. After the animal maintained its gaze on the fixation point for 300 ms, a static RDP was shown either inside the neuron's RF or opposite to the RF, cueing the animal as to the target's location in the upcoming trial. The trial was continued if the monkey touched a lever. 300 ms after the lever was touched, a second similar RDP was displayed at the other location and both RDPs began moving in the same direction. At random time points between 300 and 4000 ms after the onset of visual motion the target and distractor directions might undergo transient clockwise changes of +25° for 200 ms. In first distractor change trials, the distractor change occurred at least 600 ms before the target change. The monkey was required to detect the change in the target by releasing the lever in a reaction time window extending from 250 to 700 ms after the change onset and to ignore distractor changes to get a fluid reward. Trials were aborted without reward if the monkey deviated its gaze by more than 1° from the fixation point, failed to detect the target change within the reaction time window, responded to a distractor change, or released the lever in the absence of stimulus change. The direction change detection trials were interleaved with catch trials (5% of trials), in which no

stimulus change occurred and reward was given for maintaining fixation throughout the trial. Catch trials were excluded from all analyses.

**Data analysis**. We analyzed the electrophysiological data using custom-written scripts in MATLAB (MathWorks).

*Spike density function*. We computed SDF in each trial by convolving the spike train aligned to the onset of direction change with a Gaussian kernel of $\sigma = 10$, 20 and 40 ms at a resolution of 1 ms. The results shown in this report correspond to $\sigma = 20$ ms, except for a control analysis detailed in Supplementary Table 2.

*Neural response to a stimulus (trial-averaged response)*. For each individual MT cell in each of attentional conditions we averaged the corresponding SDFs across multiple presentations (mean ± SD = 7 ± 3) of the same visual stimulus to estimate the neural response to each of 12 stimuli as a function of time.

*Neural response averaged over time*. We computed time-averaged responses by averaging the neural responses over an analysis time window. In our analysis, we used a pre-change analysis time window from −700 to 0 ms and a post-change window from 100 to 200 ms after the direction change onset, unless otherwise stated. It has been documented that the transient response induced by change events play a key role in the sensation and perception of these rapid events[18,29,45–55]. We, thereby, considered the post-change analysis time window from 100 to 200 ms following the direction change, similar to Price and Born[56], which takes the direction change induced-transient into account. In principle, in agreement with a previous study[56], we showed that any post-change time window, which includes the transient response change yields similar results (see Supplementary Note 2: control analysis 3).

*Direction tuning curves*. We quantified the direction tuning of MT cells by fitting time-averaged responses to 12 directions with a (symmetric) von Mises function of the form:

$$y(x; a_1, a_2, a_3, a_4) = a_1 + a_2 e^{a_3 \cos(x-a_4)} \quad (1)$$

or a skewed von Mises function[57],

$$y(x; a_1, a_2, a_3, a_4, a_5) = a_1 + a_2 e^{a_3 \cos((x-a_4)+a_5(\cos(x-a_4)-1))} \quad (2)$$

The least-squares minimization routine was used to fit the data with the von Mises functions.

*Population response profile*. Attended and unattended trial-averaged responses of each neuron to different directions were normalized to the neuron's largest response evoked in either attentional condition. The normalized responses of each neuron were aligned in time to the direction change onset and in direction to the pre-change preferred direction of the neuron. The pre-change preferred direction was the median value of preferred directions determined by fitting the responses to different directions with the von Mises function every 10 ms prior to the direction change. The normalized, aligned data were averaged across the neurons to construct the population response profile for both attended (Fig. 1b) and unattended (Fig. 1c) stimuli.

We estimated the encoded stimulus direction (black solid lines in Fig. 1d–e) by least-squares fitting the population responses with the von Mises function (see Eq. 1) every 20 ms.

*Error in the location of population response curves*. To estimate the error in the location of pre- and post-change population response curves we used a bootstrapping procedure. On each iteration of the bootstrap, we took 100 random draws (with replacement) from the locations of population response curves computed every millisecond to produce a bootstrap sample. We computed the mean of 1000 bootstrap samples and computed the standard deviation of the bootstrap mean as the error in the peak location.

### Human psychophysical study

*Participants*. Twenty-one volunteers, 7 males and 14 females, aged 21–35 years, with normal or corrected-to-normal vision took part in the experiment. Nineteen were naive to the purpose of the study. The study was approved by the Ethics Committee of the Georg-Elias-Mueller Institute of Psychology of the Faculty for Biology and Psychology, University of Göttingen, and all participants signed a written consent form prior to the experiment. Ten of the subjects (three males, aged 21–35 years) fulfilled the criterion for inclusion in this study (see Procedures).

*Apparatus and stimuli*. Stimuli were programmed and controlled in an open-source software package MWorks (http://mworks-project.org/) running on a Macintosh computer. They were presented on a monitor screen Samsung (SynsMaster 2233, 1680 × 1050 pixels) in a dim room. The screen had a refresh rate of 120 Hz, a luminance of 0.1 Cd m$^{-2}$, and subtended 475 mm in width and 295 mm in height resulting in a spatial resolution of 35 pixels per degree. The observers viewed the monitor screen binocularly from a distance of 57 cm. A chin and forehead rest were

used to reduce head movements. An eye tracker with a sampling rate of 1000 Hz (EyeLink 1000, SR Research) recorded eye movements in each trial. The subjects used buttons on a gamepad (Logitech Precision) to initiate each trial and to respond at the end of each trial.

Stimuli were moving RDPs. Each RDP consisted of 100 dots (each 0.1° in diameter), all in white. The dots had a luminance of 70.2 Cd m⁻² and they were randomly plotted in a stationary circle of radius 3° of visual angle and all coherently moved in a specific direction at the speed of 8° per second.

Mask stimuli were identical, except each dot moved with the speed of 8° per second in a direction between −45° and +45° relative to the post-change motion direction (main experiment) or the motion direction (direction discrimination experiment). The centers of stimuli were at 5° eccentricity to the left and right side of the central fixation point. Data were analyzed using custom-written scripts in MATLAB (MathWorks).

*Procedures*. On each visit, subjects were given both written and verbal information about the tasks. They were also instructed to covertly attend to the right stimulus in all experiments. Suitability of the subjects for the main experiment and including their data in our analysis was based on their sensitivity in discriminating between upward and downward motion in a direction discrimination experiment: an average discrimination threshold less than 3° for both rightward and leftward motions. Ten of 21 subjects met the criterion and completed the main task. Each subject repeated each experiment three to six times on three different days. The subjects seated in front of a screen and pressed a gamepad button, while they maintained eye fixation on a white small square (subtended on each side 0.17° visual angle) presented in the middle of the screen to begin a trial. If fixation was broken at any time during the trial, the trial was aborted and repeated later.

*Motion direction discrimination task*. Two identical moving RDPs were shown for 200 ms in both visual hemifields at the same eccentricity. The motion direction in each trial was determined using simple 1-up/1-down staircase procedures for rightward and leftward motions. The two coherently moving RDPs were then replaced by the two mask RDPs. Two hundred milliseconds after the stimuli disappeared and the subjects were required to press a gamepad button to indicate whether the observed motion direction was upward or downward relative to their reference horizontal lines. Direction discrimination thresholds for each of rightward and leftward motions in each session were estimated by measuring the slope of the psychometric curve least-squares fitted with a logistic function of the form:

$$P(x; \alpha, \beta) = 1/(1 + e - \beta(x - \alpha)) \quad (3)$$

$\alpha$ is the point of subjective equality and $1/\beta$ (slope) is a measure of how accurate a subject is in judging the direction (direction threshold). The average discrimination threshold across sessions was used to determine the discrimination threshold for each subject.

*Main experiment*. Two RDPs moving in the same direction were presented in opposite hemifields at equal eccentricity. In each trial, the direction of motion was randomly chosen to be leftward or rightward. A clockwise or counterclockwise change of 22°, 25°, or 27° occurred in the direction of right RDP at a random time between 2000 and 3200 ms. Two hundred milliseconds after the change, stimuli were replaced with mask RDPs displayed for 200 ms. Subsequently, the screen went blank and subjects were required to press the buttons of a gamepad to indicate whether the motion direction after the change was upward or downward relative to their reference horizontal line. This allowed us to estimate the perceived direction change in a counterbalanced design of rightward vs. leftward motions and clockwise vs. counterclockwise direction changes. Post-change motion direction varied from trial to trial according to four simple 1-up/1-down staircase procedures associated with clockwise and counterclockwise direction changes in each of rightward and leftward motions. The post-change motion direction in each staircase began with an angle that was distinctly different from the horizontal line (20°). This angle reduced toward the horizontal line until the reported post-change direction crossed the horizontal line. The procedure was then reversed and increased the angle until the subject's perceived direction of post-change motion did not cross the horizontal line. Each staircase, therefore, yielded the 50% point of subjective equality for the corresponding motion direction and direction change. For each of rightward and leftward motion directions, the error in the perception of direction change was determined by averaging the points of subjective equality measured in the corresponding staircase procedures for clockwise and counterclockwise direction changes.

**Reporting summary**. Further information on research design is available in the Nature Research Reporting Summary linked to this article.

## Data availability
The source data underlying all figures and supplementary figures are provided as a Source Data file. Any additional data from this study are available from the corresponding authors upon reasonable request. A reporting summary for this Article is available as a Supplementary Information file.

## Code availability
The analysis code used in this study is available from the corresponding authors upon reasonable request.

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

## Acknowledgements

This work was supported by the grants of the Deutsche Forschungsgemeinschaft through the Collaborative Research Center 889 "Cellular Mechanisms of Sensory Processing" to S. T. (Project C04), the Federal Ministry of Education and Research (BMBF) of Germany under grant number 01GQ1005C. We are grateful to Takahiro Doi for helpful discussions.

## Author contributions

Conceptualization: V.M., J.M., S.T. Data curation: V.M., S.T. Formal analysis: V.M. Funding acquisition: S.T. Monkey data collection: J.M. Human data collection: V.M. Methodology: V.M., J.M., S.T. Project administration: S.T. Resources: S.T. Software: V.M. Supervision: J.M., S.T. Validation: V.M., J.M., S.T. Visualization: V.M., J.M., S.T. Writing—original draft: V.M. Writing—review and editing: V.M., J.M., S.T.

## Competing interests

The authors declare no competing interests.
