## [Peer Review File · Nature Communications]

Reviewers' comments:

Reviewer #1 (Remarks to the Author):

This manuscript by Mehrpour et al explores how attention alters the neural representation of direction changes in area MT. The authors show convincingly that attention causes a slight over-estimation of a change in motion direction, presumably enhancing detectability of the change (though this is not shown). The authors also show that attention causes a similar misestimation of direction change in human subjects.

The data are convincing. The analysis is generally straightforward (but see below). The primary limitations are: (1) The attentional effect is small since a similar but weaker over-estimation is evident in the control (no attend) condition. (2) The writing, which makes things much more confusing than need be. (3) The data description is incomplete.

Comments:

1. I found it difficult to understand the main findings from the abstract. The phrase “compromising perceptual accuracy in order to improve behavioral outcome” is opaque. The authors should be more precise and accurate, something like “compromising accurate stimulus estimation to enhance change detectability”. This is the central claim, I believe. Thus, it seems odd to refer to a loss of “perceptual accuracy” (the perception of change is enhanced). The mention of adaptation in the abstract is also confusing and not accurate. The authors explain more clearly the role of adaptation in their data in the main body but here the statement reads like a blanket assessment of attention and adaptation effects. Similar issue with final sentence of the abstract: “forgo representational accuracy” If the goal is representation of change, then accuracy is enhanced, in that the change is easier to detect. The authors mean, I think, forgo accurate representation of stimulus direction to improve encoding of stimulus change. Multiple instances of similar language in the main body should be fixed as well.

2. Please clarify how many neurons came from each monkey. And indicate whether significant and consistent results were evident in both animals. Please also provide some basic characterization of the cells' tuning: direction selectivity; and fit quality distributions for von Mises functions. Also, please provide some estimate of variance or confidence interval for the main effect, the shift shown in Fig 2a(e.g. bootstrap).

3. Fig 2: Please clarify the sign convention here. Negative shifts indicate towards/away from the pre-change direction? Also, are these shifts independent of the pre-change direction preference? This seems to be assumed in Supp Fig 4, where all of the tuning curves shift in one direction by an equal amount. Is this seen in the data? Please justify/clarify.

4. line 120: “magnitude we saw in...Figure 2B” I believe that authors mean 2A? Isn't this analysis more

closely related to the perceptual measurements?

5. Supp Info 5 is meant to assess whether there is an interaction between attention and adaptation. But the analysis shown—the relationship between attentional shifts and attention-mediated pre-change tuning parameters—doesn't really address the question of this interaction. Why not, instead, compare how much the responses decay between stimulus onset and the change (a seemingly good measure of adaptation) and the attentional shift?

Reviewer #2 (Remarks to the Author):

Manuscript summary: This manuscript proposes an alternative hypothesis concerning the mechanism by which attention improves behavioral performance. Instead of improving behavior by improving perceptual accuracy, the authors suggest that attention improves behavior by distorting the neural representation of a stimulus, which reduces perceptual accuracy. The authors report neural data from extracellular single-unit recordings from area MT while monkeys perform a motion direction change detection task, as well as psychophysical data from humans performing a motion direction discrimination task.

Overview: Prior electrophysiological studies have demonstrated that adaptation alone can reduce perceptual accuracy. The authors aim to build upon this prior work and report that the perceptual inaccuracy due to adaptation is enhanced by attention. This is potentially a conclusion of interest that has not been fully explored. However, the paper does not clearly support this claim. My main concerns are: 1) that the manuscript does not demonstrate that the effect of attention is not simply to enhance the neural effects of the release from sensory adaptation, and 2) that the authors did not demonstrate a behavioral effect of attention in either the human or the monkey data. See individual comments below.

Comments:

1. In the unattended-condition results of the electrophysiological data, the motion direction change caused a release from adaptation that resulted in an increase in neural responses and a shift in direction tuning. Attention, which increased neural responses prior to the direction change, may have simply increased that effect of adaptation. The authors spoke to this concern in the Discussion by describing Supplemental Information 5, which illustrated that there was no correlation between the effect of attention on direction tuning parameters prior to the direction change and the effect of attention on the direction-tuning shift after the direction change. This analysis essentially compared the effect of attention to itself (the effect of attention on the adapted neural responses versus the effect of attention on the release from adaptation). This analysis does not compare the effect of attention to the effect of the release from adaptation alone, which would demonstrate that the effect of attention was distinct from the effect of the release from adaptation and not an enhancement of the neural effect of this

release. To start: was there no correlation between the post-change tuning shift in the unattended condition (tuning shift due to a release from adaptation alone) and the post-change tuning shift due to attention alone (isolated by, for example, subtracting the tuning shift due to adaptation alone as calculated in the unattended condition)?

2. The human psychophysical task presented two random dot patterns to match the monkey electrophysiological task (one on the left side of the screen, one on the right side); however, the motion direction change always occurred in the right dot pattern. With a large direction change of at least 22 degrees, there was not necessarily a need for the subjects to attend to one stimulus over the other, and there were no distractors or measurements of performance in the unattended condition to demonstrate that attention was involved in the psychophysical effects observed. The authors should demonstrate that the psychophysical effects were distinct from those that would occur due to adaptation alone.

3. Just as the human psychophysical experiment did not demonstrate that attention affected behavioral performance, the monkey electrophysiological experiment did not demonstrate that attention was necessary to improve behavioral performance. This is particularly relevant because the authors hypothesize that attention compromises perceptual accuracy for the goal of improving behavioral performance. In the monkey electrophysiological experiment, detection of the 25-degree motion change to the 100% coherently moving bright dots did not necessarily require attention to improve performance beyond that which might have been achieved in the unattended condition. The authors should describe the behavioral performance in the unattended versus attended conditions.

4. It is not clear that the results were consistent across monkeys. The authors should illustrate or note the effects per individual monkey, as well as indicate the number of MT cells recorded per monkey.

Minor comments:

1. The authors should discuss the relationship between adaptation and normalization (reviewed by Solomon & Kohn, *Current Biology*, 2014) and the potential role of normalization in the relationship between adaptation and attention.

2. What was the probability of the distractor change relative to the probability of the target change? Did the timing of the target changes follow a flat hazard function? Were catch trials included?

3. What was the size of the fixation window?

Reviewers' comments:

Reviewer #1 (Remarks to the Author):

This manuscript by Mehrpour et al explores how attention alters the neural representation of direction changes in area MT. The authors show convincingly that attention causes a slight over-estimation of a change in motion direction, presumably enhancing detectability of the change (though this is not shown). The authors also show that attention causes a similar misestimation of direction change in human subjects.

The data are convincing. The analysis is generally straightforward (but see below). The primary limitations are: (1) The attentional effect is small since a similar but weaker over-estimation is evident in the control (no attend) condition. (2) The writing, which makes things much more confusing than need be. (3) The data description is incomplete.

Comments:

1. I found it difficult to understand the main findings from the abstract. The phrase “compromising perceptual accuracy in order to improve behavioral outcome” is opaque. The authors should be more precise and accurate, something like “compromising accurate stimulus estimation to enhance change detectability”. This is the central claim, I believe. Thus, it seems odd to refer to a loss of “perceptual accuracy” (the perception of change in enhanced). The mention of adaptation in the abstract is also confusing and not accurate. The authors explain more clearly the role of adaptation in their data in the main body but here the statement reads like a blanket assessment of attention and adaptation effects. Similar issue with final sentence of the abstract: “forgo representational accuracy” If the goal is representation of change, then accuracy is enhanced, in that the change is easier to detect. The authors mean, I think, forgo accurate representation of stimulus direction to improve encoding of stimulus change. Multiple instances of similar language in the main body should be fixed as well.

Thank you for this suggestion. We have clarified our language in the abstract and throughout the main body.

2. Please clarify how many neurons came from each monkey. And indicate whether significant and consistent results were evident in both animals.

We have clarified this in the Results and Methods. We have also added additional supplementary information (Supplemental Information 3):

- Results: “Our results remained consistent when parsed by the data collected from each animal individually (Supplemental Information 3).”

- Supplemental Information 3:

“Figure S13 Figure 2b replotted for each monkey individually (monkey M: 40 cells, monkey F: 12 cells). Unfilled bars represent cells for which the magnitude of tuning shift was greater than 40°. The median values (indicated by arrowheads) and the corresponding p values are listed in Table S11.”

Monkey	# cells	Attended shift (deg)	Unattended shift (deg)	Attention impact (deg)
M	40	-13 p = 0.00005	-7 p = 0.001	-6 p* = 0.03
F	12	-17 p = 0.0005	-4 p = 0.2	-13 p* = 0.01

“Table S11 The median value of direction tuning curve shifts, induced by attended and unattended direction changes of +25° for each monkey individually. p is the p-value of the two-sided Wilcoxon signed rank test for distributions with zero median. p* is the p-value of the paired two-sided Wilcoxon signed rank test for the null hypothesis that the difference between paired samples comes from a distribution with zero median.”

- **Methods:** “Extracellular single-unit recordings were made from two hemispheres of two awake, behaving monkeys seated in a primate chair with the head restrained. The neural responses were recorded from 52 isolated MT cells (40 neurons from monkey M and 12 from monkey F) with tungsten microelectrodes (impedance 0.5–2 Ω, Microprobe and FHC).”

Please also provide some basic characterization of the cells’ tuning: direction selectivity; and fit quality distributions for von Mises functions.

As suggested by the reviewer, we have now included the distributions of coefficient of determination (R^2) and directionality index for von Mises functions across neurons and conditions in Supplemental Information 2c-d:

- **Supplemental Information 2c-d:**

“(c) Distribution of coefficient of determination (R^2) for the von Mises function across cells and conditions ($n = 208 = 52 \text{ cells} \times 4 \text{ conditions per cell}$). The median value is marked. (d) Distribution of directionality index, $DI = (\text{Max Response} - \text{Min Response}) / (\text{Max Response} + \text{Min Response})$, for the von Mises function across cells and conditions ($n = 208$). The median value is marked.”

Also, please provide some estimate of variance or confidence interval for the main effect, the shift shown in Fig 2a(e.g. bootstrap).

This is now clarified in the legend of Figure 2a and Methods:

- **Figure 2a:** “Error in the peak location obtained from the bootstrap procedure was 0.1° for pre-change (attended), 0.6° for post-change (attended), 0.2° for pre-change (unattended), and 0.4° for post-change (unattended) (see Methods for details).”

- **Methods:** “Error in the location of population response curves: To estimate the error in the location of pre- and post-change population response curves we used a bootstrapping procedure. On each iteration of the bootstrap, we took 100 random draws (with replacement) from the locations of population response curves computed every millisecond to produce a bootstrap sample. We computed the mean of 1,000 bootstrap samples and computed the standard deviation of the bootstrap mean as the error in the peak location.”

3. Fig 2: Please clarify the sign convention here. Negative shifts indicate towards/away from the pre-change direction?

We apologize for the lack of clarity here. We have clarified the sign convention in Figure 2 and Supplemental Information 2a.

According to the sign convention used, a clockwise (positive) direction change induces a negative shift in the direction tuning curve (i.e. a counterclockwise displacement of the direction tuning curve). In other words, a negative shift indicates that the pre-change preferred direction shifts away from the pre-change directions on the right side of the tuning peak, while the pre-change preferred direction moves towards the pre-change directions on the left side of the peak.

- **Figure 2:** “The sign convention is that positive and negative values denote clockwise and counterclockwise displacements, respectively (Supplemental Information 2a).”

- **Figure SI2a:**

“Illustration of motion directions and sign convention in both polar (top) and Cartesian (bottom) coordinate systems. Positive and negative signs denote clockwise (rightward) and counterclockwise (leftward) displacements, respectively.”

Also, are these shifts independent of the pre-change direction preference? This seems to be assumed in Supp Fig 4, where all of the tuning curves shift in one direction by an equal amount. Is this seen in the data? Please justify/clarify.

We observed a weak correlation between the pre-change preferred directions of MT cells and direction tuning shifts induced by the direction change (Pearson $r = 0.27$, $p = 0.053$ for attended; and Pearson $r = 0.08$, $p = 0.6$ for unattended. Spearman $r = 0.19$, $p = 0.2$ for attended; and Spearman $r = 0.12$, $p = 0.4$ for unattended):

Figure: Relationship between direction tuning shifts and cells' preferred directions ($n = 52$ cells \times 2 conditions per cell = 104). Blue and red circles represent attended and unattended conditions, respectively. The Pearson correlation coefficients are labeled. Data points corresponding to the tuning shifts with magnitudes greater than 40° are not shown (attended: 6 data points, unattended: 5 data points).

We have clarified this point in the Supplemental Information 5:

- Supplemental Information 5: “[...] This is justified because there was a weak correlation between the pre-change preferred directions of MT cells and the direction tuning shifts induced by the direction change (Pearson $r = 0.27$, $p = 0.053$ for attended; and Pearson $r = 0.08$, $p = 0.6$ for unattended. Spearman $r = 0.19$, $p = 0.2$ for attended; and Spearman $r = 0.12$, $p = 0.4$ for unattended).”

4. line 120: “magnitude we saw in...Figure 2B” I believe that authors mean 2A? Isn't this analysis more closely related to the perceptual measurements?

The reviewer is correct. The magnitude of perceptual effects is more closely related to those shown in Figure 2A. This correction has now been made.

5. Supp Info 5 is meant to assess whether there is an interaction between attention and adaptation. But the analysis shown—the relationship between attentional shifts and attention-mediated pre-change tuning parameters—doesn't really address the question of this interaction. Why not, instead, compare how much the responses decay between stimulus onset and the change (a seemingly good measure of adaptation) and the attentional shift?

Thank you for the suggestion. The idea behind the analysis we previously reported in the Supplemental Information 5 (now Supplemental Information 6) was that if the increase in the tuning shifts associated with attention is caused by the attentional modulation of responses prior to the direction change, we would expect that the neurons with larger attentional modulation of their responses exhibit larger attentional shifts. This is clarified in the Discussion. In addition to this, based on a new set of analyses we have now added the graph shown below into Supplemental Information 6. Figure SI6a illustrates the relationship between the decay of responses over the course of visual motion exposure prior to the direction change and the attentional shifts. The data show that there is no correlation between the responses decay over visual motion exposure prior to the direction change and the attentional shifts (Pearson $r = 0.04$, $p = 0.8$). We also discuss this in Discussion now.

- Discussion: “Could an increased adaptation, caused by the attentional increase of responses cause this effect? Our data argue against this. If the increase in the tuning shifts associated with attention (attentional shifts, attended shift - unattended shift) is caused by the attentional modulation

of pre-change responses, we would expect that neurons with larger attentional modulation of their responses exhibit larger attentional shifts. Instead, we found that there is no correlation between the attentional modulation of pre-change tuning parameters and the attentional shifts (Pearson correlation: $r < 0.2$, $p > 0.05$). To further investigate the interaction between attention and pre-change visual motion adaptation, we examined the relationship between the decay of responses over the course of visual motion exposure prior to the direction change as a measure of adaptation and the attentional shift and did not find any correlation between them (Pearson correlation: $r = 0.04$, $p = 0.8$; **Fig. S16a**).

- **Figure S16a:**

Figure S16: Interaction between attention and adaptation in the neural representation of direction changes. (a) The attentional shift (i.e. attended shift – unattended shift) is plotted against the visual motion adaptation assessed by the average adaptation index across attentional conditions ($n = 52$ cells). The adaptation index for each attentional condition is defined as $(R_{\text{pre-change}} - R_{\text{post-onset}}) / (R_{\text{pre-change}} + R_{\text{post-onset}})$, where $R_{\text{pre-change}}$ and $R_{\text{post-onset}}$ are the cell's average responses (across directions) to the pre-change (300 ms prior to the direction change) and post-onset (from 400 to 700 ms) stimuli. Data points corresponding to the attentional shifts with magnitudes greater than 40° are not shown (6 data points)

Reviewer #2 (Remarks to the Author):

Manuscript summary: This manuscript proposes an alternative hypothesis concerning the mechanism by which attention improves behavioral performance. Instead of improving behavior by improving perceptual accuracy, the authors suggest that attention improves behavior by distorting the neural representation of a stimulus, which reduces perceptual accuracy. The authors report neural data from extracellular single-unit recordings from area MT while monkeys perform a motion direction change detection task, as well as psychophysical data from humans performing a motion direction discrimination task.

Overview: Prior electrophysiological studies have demonstrated that adaptation alone can reduce perceptual accuracy. The authors aim to build upon this prior work and report that the perceptual inaccuracy due to adaptation is enhanced by attention. This is potentially a conclusion of interest that has not been fully explored. However, the paper does not clearly support this claim. My main concerns are: 1) that the manuscript does not demonstrate that the effect of attention is not simply to enhance the neural effects of the release from sensory adaptation, and 2) that the authors did not demonstrate a behavioral effect of attention in either the human or the monkey data. See individual comments below.

Comments:

1. In the unattended-condition results of the electrophysiological data, the motion direction change caused a release from adaptation that resulted in an increase in neural responses and a shift in direction tuning. Attention, which increased neural responses prior to the direction change, may have simply increased that effect of adaptation. The authors spoke to this concern in the Discussion by describing Supplemental Information 5, which illustrated that there was no correlation between the effect of attention on direction tuning parameters prior to the direction change and the effect of attention on the direction-tuning shift after the direction change. This analysis essentially compared the effect of attention to itself (the effect of attention on the adapted neural responses versus the effect of attention on the release from adaptation).

The idea behind the analysis previously reported in the Supplemental Information 5 (now Supplemental Information 6) is that if the increase in the tuning shifts associated with attention is caused by the attentional modulation of responses prior to the direction change, we would expect that the neurons with larger attentional modulation of their responses exhibit larger attentional shifts. This is now clarified in the Discussion. Based on a new set of analyses we have added a plot in the Supplemental Information 6 (see below).

- Discussion: "Could an increased adaptation, caused by the attentional increase of responses cause this effect? Our data argue against this. If the increase in the tuning shifts associated with attention (attentional shifts, attended shift - unattended shift) is caused by the attentional modulation of pre-change responses, we would expect that the neurons with larger attentional modulation of their responses exhibit larger attentional shifts. Instead, we found that there is no correlation between the attentional modulation of pre-change tuning parameters and the attentional shifts (Pearson correlation: $r < 0.2$, $p > 0.05$)."

This analysis does not compare the effect of attention to the effect of the release from adaptation alone, which would demonstrate that the effect of attention was distinct from the effect of the release from adaptation and not an enhancement of the neural effect of this release. To start: was there no correlation between the post-change tuning shift in the unattended condition (tuning shift due to a release from adaptation alone) and the post-change tuning shift due to attention alone (isolated by, for example, subtracting the tuning shift due to adaptation alone as calculated in the unattended condition)?

We highlight the main finding of our study before we discuss the interaction between attention and adaptation here. Our results reveal that a positive (rightward) attended direction change induces (in average) a negative (leftward) shift in the direction tuning curves of MT units (see Figure 2b, top panel for the distribution of direction tunings' shifts in attended condition). The same but behaviorally irrelevant (unattended) direction change causes smaller negative tuning curve shifts (see Figure 2b,

bottom panel for the distribution of direction tunings' shifts in unattended condition). As suggested by the reviewer, if attention increases the adaptation-induced tuning shifts simply by modulating the pre-change responses, we would expect that the units with larger negative sensory shifts in unattended condition (which presumably underwent stronger adaptation over the course of pre-change visual motion exposure) exhibit larger negative shifts due to attention alone (i.e. isolated effect of attention = attended shift – unattended shift. We will refer to this as *attentional shift*). In other words, there should be a positive correlation between the shift in sensory condition (unattended) and the attentional shift. Our analysis (Supplemental Information 6) shows the opposite relationship (negative correlation, Pearson $r = -0.81$, $p=2 \times 10^{-13}$): units with large negative tuning shifts in unattended condition were those either not affected by attention, or even showed positive attentional shifts (opposite to the sign of the main effect) and vice versa (i.e. the units with positive sensory shifts had the largest negative attentional shifts). This suggests that attention does not increase the sensory shifts simply by modulating the pre-change responses, but rather combines with the effect of adaptation to improve the neural representation of the change event. This is now included in the Discussion and Supplemental Information 6.

- Discussion: “[...] if attention increases the adaptation-induced tuning shifts simply by modulating the pre-change responses, we would expect that the cells with larger negative sensory shifts in unattended condition (which presumably underwent stronger adaptation over the course of pre-change visual motion exposure) exhibit larger negative attentional shifts. In other words, there should be a positive correlation between the shift in sensory condition and the attentional shift. Our analysis (Fig. SI6b) shows an opposite relationship (negative correlation, Pearson $r = -0.81$, $p = 2 \times 10^{-13}$): cells with large negative tuning shifts in unattended condition were those either not affected by attention, or even showed positive attentional shifts (opposite to the sign of main effect) and vice versa (i.e. the cells with positive sensory shifts had the largest negative attentional shifts). This suggests that attention does not increase the sensory shifts simply by modulating the pre-change responses, but rather combines with the effect of adaptation to improve the neural representation of the change event.”

- Figure SI6b:

“Relationship between attentional shift (attended shift – unattended shift), plotted along the y-axis and the sensory (unattended) shift represented along the x-axis (n = 52 cells). Data points corresponding to the shifts with magnitudes greater than 40° are not shown (9 data points). The Pearson correlation coefficients are labeled on each graph.”

2. The human psychophysical task presented two random dot patterns to match the monkey electrophysiological task (one on the left side of the screen, one on the right side); however, the motion direction change always occurred in the right dot pattern. With a large direction change of at least 22 degrees, there was not necessarily a need for the subjects to attend to one stimulus over the other, and there were no distractors or measurements of performance in the unattended condition to demonstrate that attention was involved in the psychophysical effects observed. The authors should demonstrate that the psychophysical effects were distinct from those that would occur due to adaptation alone.

Thank you for pointing this out. We agree that a different paradigm would have been necessary to isolate the effects of attention on the perception of behaviorally relevant direction changes. However,

this was not the goal of our psychophysical study. Rather, we aimed to demonstrate that assuming a similar neural misrepresentation in humans as the one we observed in monkeys would lead to a perceptual overestimation.

Given that the subjects had to perform a *discrimination* task on the post-change motion direction (where a staircase kept the post-change directions close to the subjects' perceptual thresholds), the task was quite challenging and subjects reported that they needed to maximally attend to the cued (right) stimulus to perform well. As in our experiment subjects were not required to detect the direction changes (but rather to discriminate the post-change motion directions), the direction change magnitude in our task could not help the subjects to discriminate the post-change directions.

We have now clarified these points in the Results.

- Results: "Note, that it was not our intention to isolate the effect of attention on the perception of behaviorally relevant direction changes. Rather, we aimed to demonstrate that (assuming a similar neural misrepresentation in humans and our monkeys) humans experience a perceptual overestimation of the magnitude of attended direction changes."

3. Just as the human psychophysical experiment did not demonstrate that attention affected behavioral performance, the monkey electrophysiological experiment did not demonstrate that attention was necessary to improve behavioral performance. This is particularly relevant because the authors hypothesize that attention compromises perceptual accuracy for the goal of improving behavioral performance. In the monkey electrophysiological experiment, detection of the 25-degree motion change to the 100% coherently moving bright dots did not necessarily require attention to improve performance beyond that which might have been achieved in the unattended condition. The authors should describe the behavioral performance in the unattended versus attended conditions.

In paradigms, such as ours, that involve attended and fully unattended stimuli, a comparison of behavioral performance in the attended versus unattended conditions is not possible, as behavioral performance requires at least some attentional allocation to a stimulus. Nevertheless, it is clear that the monkeys in our task followed the instructions (i.e. the cue) and selectively attended to the cued stimulus, as the difference we observe (here and in multiple other studies using this paradigm) in the neuronal responses to the cued vs. the uncued stimulus differ significantly (e.g. see Supplemental Information 1). One contributing factor to this selective allocation of spatial attention is that the task is more challenging than one might imagine, presumably because the stimulus change is very brief and the following time window for the animal's response is also short. Together these parameters ensure that the animals' performance is high, but far from saturated. We have now clarified this and included the behavioral performance values in the Results.

Beyond the scope of this manuscript we examined the relationship between the monkeys' behavioral performance in detecting the direction changes and the level of attention the monkeys allocated to the stimulus. This analysis does not directly relate to the current manuscript and is thus intended for publication elsewhere. A summary of this analysis is appended to the end of our response letter.

- Results: "To evaluate the effects of attention on the accuracy of the neural representation of the magnitude of abrupt changes we used a well-established spatial attention paradigm^{6,7,24,25}."

- Results: "The monkeys were rewarded for detecting a direction change of the cued stimulus within a reaction time window from 250 to 700 ms after the change onset (**Fig. 1a**; see Methods for details). Given that the stimulus change is brief and the following reaction time window is also short, the task is challenging, with the animals' performance high, but far from saturated ($87\% \pm 1\%$ for monkey M; $87\% \pm 2\%$ for monkey F)."

4. It is not clear that the results were consistent across monkeys. The authors should illustrate or note the effects per individual monkey, as well as indicate the number of MT cells recorded per monkey.

We have clarified this in the Results and Methods. We have also added additional supplementary information (Supplemental Information 3):

- **Results:** “Our results remained consistent when parsed by the data collected from each animal individually (Supplemental Information 3).”

- **Supplemental Information 3:**

“**Figure S13 Figure 2b** replotted for each monkey individually (monkey M: 40 cells, monkey F: 12 cells). Unfilled bars represent cells for which the magnitude of tuning shift was greater than 40°. The median values (indicated by arrowheads) and the corresponding p values are listed in **Table S11**.”

Monkey	# cells	Attended shift (deg)	Unattended shift (deg)	Attention impact (deg)
M	40	-13 p = 0.00005	-7 p = 0.001	-6 p* = 0.03
F	12	-17 p = 0.0005	-4 p = 0.2	-13 p* = 0.01

“**Table S11** The median value of direction tuning curve shifts, induced by attended and unattended direction changes of +25° for each monkey individually. p is the p-value of the two-sided Wilcoxon signed rank test for distributions with zero median. p* is the p-value of the paired two-sided Wilcoxon signed rank test for the null hypothesis that the difference between paired samples comes from a distribution with zero median.”

- **Methods:** “Extracellular single-unit recordings were made from two hemispheres of two awake, behaving monkeys seated in a primate chair with the head restrained. The neural responses were recorded from 52 isolated MT cells (40 neurons from monkey M and 12 from monkey F) with tungsten microelectrodes (impedance 0.5–2 Ω, Microprobe and FHC).”

Minor comments:

1. The authors should discuss the relationship between adaptation and normalization (reviewed by Solomon & Kohn, Current Biology, 2014) and the potential role of normalization in the relationship between adaptation and attention.

Thank you for this suggestion. In the revision, we discuss this point in the Discussion:

Discussion: “Normalization mechanisms have been used to explain the effects of adaptation³⁸ and attention^{39,40} and they offer a plausible explanation for our electrophysiological findings. These mechanisms assume that visual motion exposure not only suppresses the feedforward drive of neurons but also weakens the activity of the neurons that make up the normalization pools, resulting in a decrease in the suppressive drive and therefore response facilitation³⁸. A normalization model that incorporates both adaptation and attention might be able to account for the effects induced by unattended and attended direction changes. Based on this model, direction change overestimation in MT results from the suppression of the feedforward drive as well as a weakened normalization following the visual motion exposure. Attention might increase the overestimation by modulating both components.”

2. What was the probability of the distractor change relative to the probability of the target change?

The probabilities of (first) distractor change and (first) target change in our electrophysiological experiment were about 0.40 and 0.60, respectively. Therefore, the ratio of the probability of the distractor change to the probability of the target change was about 0.65. Both monkeys performed the task very well such that the mean and standard error (across experimental sessions) of monkeys' behavioral performances on the direction change detection task were $87\% \pm 1\%$ for monkey M (40 sessions) and $87\% \pm 2\%$ for monkey F (12 sessions). We have added these details to the Results: "**Results:** The monkeys were rewarded for detecting a direction change of the cued stimulus within a reaction time window from 250 to 700 ms after the change onset (**Fig. 1a**; see Methods for details). Given that the stimulus change is brief and the following reaction time window is also short, the task is challenging, with the animals' performance high, but far from saturated ($87\% \pm 1\%$ for monkey M; $87\% \pm 2\%$ for monkey F)."

Did the timing of the target changes follow a flat hazard function?

As discussed in 'Appendix for reviewer 2', target direction change time in our experiment was sampled from a bimodal probability distribution throughout the recording sessions (**Figure a, c, e**). The hazard function and the subjective hazard function (anticipation function) associated with this probability distribution function are illustrated in **Figure b, d, f**.

Were catch trials included?

Yes, they were. We interleaved catch trials (a total of 5% of trials), in which no stimulus change occurred and the monkeys were rewarded for maintaining fixation. Catch trials were excluded from all analyses. We have incorporated these details into the Methods:

- **Methods:** "The direction change detection trials were interleaved with catch trials (5% of trials), in which no stimulus change occurred and reward was given for maintaining fixation throughout the trial. Catch trials were excluded from all analyses."

3. What was the size of the fixation window?

The radius of the fixation window was 1° . This is now mentioned in the Methods:

- **Methods:** "Trials were aborted without reward if the monkey deviated its gaze by more than 1° from the fixation point, [...]"

Appendix for reviewer 2

Beyond the scope of this manuscript we examined the relationship between the monkeys' behavioral performance in detecting the direction changes and the level of attention the monkeys allocated to the stimulus^{1,2}. This analysis does not directly relate to the current manuscript and is thus intended for publication elsewhere. Nevertheless, it might be of interest to reviewer 2 and is therefore presented below.

In our experiment, the direction changes time was a random variable with a bimodal probability distribution (schedule) throughout the recording sessions (**Figure a**, gray histogram plotted on the left axis). This schedule allows the monkeys to anticipate the probability of an upcoming behaviorally relevant direct change event and efficiently allocate attention at the right time to detect the changes. The temporal evolution of such anticipation can be described by a hazard function (**Figure a**, blue line plotted on the right axis). As elapsed time cannot be precisely measured by the brain^{1,3}, we replaced the hazard rate by the subjective hazard rate or anticipation function indicating that the monkeys know the elapsed time with uncertainty that scales with time (**Figure b**, blue line plotted on the right axis). **Figure b** shows that the anticipation function is concave with a peak around 2.2 s, suggesting that monkeys allocated their highest level of attention around this time while paying less attention earlier and later. To evaluate behavioral performance, we measured the monkeys' reaction times for detecting the direction changes for all trials of different waiting times (**Figure b**, black line plotted on the left axis). As shown in **Figure b** and in line with a previous work¹, the behavioral reaction time is a convex function with a trough around 2.2 s when the anticipation function peaked (i.e., when the animals allocated their highest level of attention to the target). This inverse relationship between anticipation function and reaction times (**Figure b**, blue and black lines) demonstrates that allocating more attention to the direction change event in our task systematically improved the behavioral performance. These findings were true for each monkey individually (**Figure c-f**). See Ref. 1 for details about methods.

Appendix references

1. Janssen, P. & Shadlen, M. N. A representation of the hazard rate of elapsed time in macaque area LIP. *Nat. Neurosci.* **8**, 234–41 (2005).
2. Ghose, G. M. & Maunsell, J. H. R. Attentional modulation in visual cortex depends on task timing. *Nature* **419**, 616–20 (2002).
3. Gallistel, C. R. & Gibbon, J. Time, rate, and conditioning. *Psychol. Rev.* **107**, 289–344 (2000).

Figure Behavioral effect of attention on detecting the direction changes. (a, c, e) Distribution of direction change times (grey histogram plotted on the left y-axis) and the corresponding hazard function (blue line plotted on the right y-axis) for (a) the pooled data from both monkeys (24,254 trials), and (c, e) individual animals (monkey M: 20,360 trials, monkey F: 3,894 trials). (b, d, f) The subjective hazard rate or anticipation function (blue line plotted on the right axis) and the behavioral reaction time (black line plotted on the left y-axis) for (b) the pooled data from both monkeys and (d, f) individual animals. In the case of reaction time, we split the target direction change times into five evenly spaced quantiles and computed the monkeys' reaction time for each quantile. Gray circles indicate mean values and horizontal and vertical error bars are the standard error of the mean values.

****REVIEWERS' COMMENTS:**

Reviewer #1 (Remarks to the Author):

The authors have addressed the majority of my concerns. The discussion of adaptation is much improved and the new analyses helpful. I would suggest though not to state (lines 212-213) “an attentional enhancement of the perceptual consequences of adaptation”. This implies that if there is no adaptation, there will be no attentional distortion. This may or may not be the case; the authors data don't speak to this. So perhaps: “attention, like adaptation, causes an over-representation of motion change”.

Also, at some points (e.g. lines 174-180), the authors' language may lead readers to think the human perceptual experiments show effects consistent with the attentional contribution to motion change over-estimation (not the case) rather than that the over-estimation that is present also in unattended conditions (i.e. due to adaptation). It would be helpful to add a sentence in the discussion stating clearly that the perceptual experiment, while invoking attention, don't actually show that the effects measured are due to attention (rather than, for example, adaptation).

Reviewer #2 (Remarks to the Author):

The revised manuscript has addressed all of my prior concerns. I recommend the revised manuscript for publication.

Reviewers' comments:

Reviewer #1 (Remarks to the Author):

The authors have addressed the majority of my concerns. The discussion of adaptation is much improved and the new analyses helpful. I would suggest though not to state (lines 212-213) “an attentional enhancement of the perceptual consequences of adaptation”. This implies that if there is no adaptation, there will be no attentional distortion. This may or may not be the case; the authors data don't speak to this. So perhaps: “attention, like adaptation, causes an over-representation of motion change”.

Thank you for this suggestion. We have clarified this in our manuscript:

“All together, these results demonstrate that while adaptation might cause the misrepresentation in the unattended condition, the larger effects in the attended condition are not the direct results of an attentionally enhanced adaptation, but rather reflect attention, like adaptation, causes an overrepresentation of motion change.”

Also, at some points (e.g. lines 174-180), the authors' language may lead readers to think the human perceptual experiments show effects consistent with the attentional contribution to motion change over-estimation (not the case) rather than that the over-estimation that is present also in unattended conditions (i.e. due to adaptation). It would be helpful to add a sentence in the discussion stating clearly that the perceptual experiment, while invoking attention, don't actually show that the effects measured are due to attention (rather than, for example, adaptation).

Thank you for pointing this out. We agree with your suggestion and have added a sentence to address this concern:

“Note that our human psychophysical experiment, while invoking attention, does not distinguish between the effects of attention and adaptation.”

Reviewer #2 (Remarks to the Author):

The revised manuscript has addressed all of my prior concerns. I recommend the revised manuscript for publication.